# UCB EXPLORATION VIA $Q$-ENSEMBLES

## ABSTRACT

We show how an ensemble of $Q^*$-functions can be leveraged for more effective exploration in deep reinforcement learning. We build on well established algorithms from the bandit setting, and adapt them to the $Q$-learning setting. We propose an exploration strategy based on upper-confidence bounds (UCB). Our experiments show significant gains on the Atari benchmark.

## 1 INTRODUCTION

Deep reinforcement learning seeks to learn mappings from high-dimensional observations to actions. Deep $Q$-learning (Mnih et al. (2015)) is a leading technique that has been used successfully, especially for video game benchmarks. However, fundamental challenges remain, for example, improving sample efficiency and ensuring convergence to high quality solutions. Provably optimal solutions exist in the bandit setting and for small MDPs, and at the core of these solutions are exploration schemes. However these provably optimal exploration techniques do not extend to deep RL in a straightforward way.

Bootstrapped DQN (Osband et al. (2016)) is a previous attempt at adapting a theoretically verified approach to deep RL. In particular, it draws inspiration from *posterior sampling for reinforcement learning* (PSRL, Osband et al. (2013); Osband and Van Roy (2016)), which has near-optimal regret bounds. PSRL samples an MDP from its posterior each episode and exactly solves $Q^*$, its optimal $Q$-function. However, in high-dimensional settings, both approximating the posterior over MDPs and solving the sampled MDP are intractable. Bootstrapped DQN avoids having to establish and sample from the posterior over MDPs by instead approximating the posterior over $Q^*$. In addition, bootstrapped DQN uses a multi-headed neural network to represent the $Q$-ensemble. While the authors proposed bootstrapping to estimate the posterior distribution, their empirical findings show best performance is attained by simply relying on different initializations for the different heads, not requiring the sampling-with-replacement process that is prescribed by bootstrapping.

In this paper, we design new algorithms that build on the $Q$-ensemble approach from Osband et al. (2016). However, instead of using posterior sampling for exploration, we construct uncertainty estimates from the $Q$-ensemble. Specifically, we first propose the Ensemble Voting algorithm where the agent takes action by a majority vote from the $Q$-ensemble. Next, we propose the UCB exploration strategy. This strategy is inspired by established UCB algorithms in the bandit setting and constructs uncertainty estimates of the $Q$-values. In this strategy, agents are optimistic and take actions with the highest UCB. We demonstrate that our algorithms significantly improve performance on the Atari benchmark.

## 2 BACKGROUND

### 2.1 NOTATION

We model reinforcement learning as a Markov decision process (MDP). We define an MDP as $(\mathcal{S}, \mathcal{A}, T, R, p_0, \gamma)$, in which both the state space $\mathcal{S}$ and action space $\mathcal{A}$ are discrete, $T : \mathcal{S} \times \mathcal{A} \times \mathcal{S} \mapsto \mathbb{R}_+$ is the transition distribution, $R : \mathcal{S} \times \mathcal{A} \mapsto \mathbb{R}$ is the reward function, assumed deterministic given the state and action, and $\gamma \in (0, 1]$ is a discount factor, and $p_0$ is the initial state distribution. We denote a transition experience as $\tau = (s, a, r, s')$ where $s' \sim T(s'|s, a)$ and $r = R(s, a)$. A policy $\pi : \mathcal{S} \mapsto \mathcal{A}$ specifies the action taken after observing a state. We denote the $Q$-function for policy $\pi$ as $Q^\pi(s, a) := \mathbb{E}_\pi \left[ \sum_{t=0}^\infty \gamma^t r_t | s_0 = s, a_0 = a \right]$ where $r_t = R(s_t, a_t)$. The optimal $Q^*$-function

corresponds to taking the optimal policy

$$Q^*(s, a) := \sup_\pi Q^\pi(s, a)$$

and satisfies the Bellman equation

$$Q^*(s, a) = \mathbb{E}_{s' \sim T(\cdot|s,a)} \big[ r + \gamma \cdot \max_{a'} Q^*(s', a') \big].$$

## 2.2 EXPLORATION IN REINFORCEMENT LEARNING

A notable early optimality result in reinforcement learning was the proof by Watkins and Dayan Watkins (1989); Watkins and Dayan (1992) that an online $Q$-learning algorithm is guaranteed to converge to the optimal policy, provided that every state is visited an infinite number of times. However, the convergence of Watkins' Q-learning can be prohibitively slow in MDPs where $\epsilon$-greedy action selection explores state space randomly. Later work developed reinforcement learning algorithms with provably fast (polynomial-time) convergence (Kearns and Singh (2002); Brafman and Tennenholtz (2002); Strehl et al. (2006)). At the core of these provably-optimal learning methods is some exploration strategy, which actively encourages the agent to visit novel state-action pairs. For example, R-MAX optimistically assumes that infrequently-visited states provide maximal reward, and delayed $Q$-learning initializes the $Q$-function with high values to ensure that each state-action is chosen enough times to drive the value down.

Since the theoretically sound RL algorithms are not computationally practical in the deep RL setting, deep RL implementations often use simple exploration methods such as $\epsilon$-greedy and Boltzmann exploration, which are often sample-inefficient and fail to find good policies. One common approach of exploration in deep RL is to construct an exploration bonus, which adds a reward for visiting state-action pairs that are deemed to be novel or informative. In particular, several prior methods define an exploration bonus based on a density model or dynamics model. Examples include VIME by Houthooft et al. (2016), which uses variational inference on the forward-dynamics model, and Tang et al. (2016), Bellemare et al. (2016), Ostrovski et al. (2017), Fu et al. (2017). While these methods yield successful exploration in some problems, a major drawback is that this exploration bonus does not depend on the rewards, so the exploration may focus on irrelevant aspects of the environment, which are unrelated to reward.

## 2.3 BAYESIAN REINFORCEMENT LEARNING

Earlier works on Bayesian reinforcement learning include Dearden et al. (1998; 1999). Dearden et al. (1998) studied Bayesian $Q$-learning in the model-free setting and learned the distribution of $Q^*$-values through Bayesian updates. The prior and posterior specification relied on several simplifying assumptions, some of which are not compatible with the MDP setting. Dearden et al. (1999) took a model-based approach that updates the posterior distribution of the MDP. The algorithm samples from the MDP posterior multiple times and solving the $Q^*$ values at every step. This approach is only feasible for RL problems with very small state space and action space. Strens (2000) proposed posterior sampling for reinforcement learning (PSRL). PSRL instead takes a single sample of the MDP from the posterior in each episode and solves the $Q^*$ values. Recent works including Osband et al. (2013) and Osband and Van Roy (2016) established near-optimal Bayesian regret bounds for episodic RL. Sorg et al. (2012) models the environment and constructs exploration bonus from variance of model parameters. These methods are experimented on low dimensional problems only, because the computational cost of these methods is intractable for high dimensional RL.

## 2.4 BOOTSTRAPPED DQN

Inspired by PSRL, but wanting to reduce computational cost, prior work developed approximate methods. Osband et al. (2014) proposed randomized least-square value iteration for linearly-parameterized value functions. Bootstrapped DQN Osband et al. (2016) applies to $Q$-functions parameterized by deep neural networks. Bootstrapped DQN (Osband et al. (2016)) maintains a $Q$-ensemble, represented by a multi-head neural net structure to parameterize $K \in \mathbb{N}_+$ $Q$-functions. This multi-head structure shares the convolution layers but includes multiple "heads", each of which defines a $Q$-function $Q_k$.

Bootstrapped DQN diversifies the $Q$-ensemble through two mechanisms. The first mechanism is independent initialization. The second mechanism applies different samples to train each $Q$-function.

These $Q$-functions can be trained simultaneously by combining their loss functions with the help of a random mask $m_\tau \in \mathbb{R}_+^K$

$$L = \sum\nolimits_{\tau \in B_{\text{mini}}} \sum\nolimits_{k=1}^K m_\tau^k \cdot (Q^k(s, a; \theta) - y_\tau^{Q_k})^2,$$

where $y_\tau^{Q_k}$ is the target of the $k$th $Q$-function. Thus, the transition $\tau$ updates $Q_k$ only if $m_\tau^k$ is nonzero. To avoid the overestimation issue in DQN, bootstrapped DQN calculates the target value $y_\tau^{Q_k}$ using the approach of Double DQN (Van Hasselt et al. (2016)), such that the current $Q_k(\cdot; \theta_t)$ network determines the optimal action and the target network $Q_k(\cdot; \theta^-)$ estimates the value

$$y_\tau^{Q_k} = r + \gamma \max_a Q^k(s', \operatorname*{argmax}_a Q_k(s', a; \theta_t); \theta^-).$$

In their experiments on Atari games, Osband et al. (2016) set the mask $m_\tau = (1, \ldots, 1)$ such that all $\{Q_k\}$ are trained with the same samples and their only difference is initialization. Bootstrapped DQN picks one $Q_k$ uniformly at random at the start of an episode and follows the greedy action $a_t = \operatorname{argmax}_a Q_k(s_t, a)$ for the whole episode.

## 3 ENSEMBLE VOTING

Ignoring computational costs, the ideal Bayesian approach to reinforcement learning is to maintain a posterior over the MDP. However, with limited computation and model capacity, it is more tractable to maintain a posterior of the $Q^*$-function. This motivates using a $Q$-ensemble as a particle filter-based approach to approximate the posterior over $Q^*$-function and we display our first proposed method, Ensemble Voting, in Algorithm 1.

Each $Q_k$ in the $Q$-ensemble $\{Q_k\}_{k=1}^K$ is parametrized with a deep neural network whose parameters are initialized independently at the start of training. Each $Q_k$ proposes an action that maximizes the $Q$-value according to $Q_k$ at every time step and the agent chooses the action by a majority vote

$$a_t = \text{MajorityVote}(\{\operatorname*{argmax}_a Q_k(s_t, a)\}_{k=1}^K).$$

At each learning interval, a minibatch of transitions is sampled from the replay buffer and each $Q_k$ takes a Bellman update based on this minibatch. For stability, Algorithm 1 also uses a target network for each $Q_k$ as in Double DQN in the batched update.

We point out that the difference among the parameters of the $Q$-ensemble $\{Q_k\}$ comes only from the independent random initialization. The deep neural network parametrization of the $Q$-ensemble introduces nonconvexity into the objective function of Bellman update, so the $Q$-ensemble $\{Q_k\}$ do not converge to the same $Q$-function during training even though they are trained with the same minibatches at every update. We also experimented with bagging by updating each $Q_k$ using an independently drawn minibatch. However, bagging led to inferior learning performance. This phenomenon that that bagging deteriorates the performance of deep ensembles is also observed in supervised learning settings. Lee et al. (2015) observed that supervised learning trained with deep ensembles with random initializations perform better than bagging for deep ensembles. Lakshminarayanan et al. (2016) used deep ensembles for uncertainty estimates and also observed that bagging deteriorated performance in their experiments.

Lu and Van Roy (2017) develop ensemble sampling for bandit problems with deep neural network parametrized policies and the theoretical justification. We derive a posterior update rule for the $Q^*$ function and approximations to the posterior update using ensembles in Appendix C. We note that in bootstrapped DQN, ensemble voting is applied for evaluation while Algorithm 1 uses ensemble voting during learning. In the experiments (Sec. 5), we demonstrate that Algorithm 1 is superior to bootstrapped DQN. The action choice of Algorithm 1 is exploitation only. In the next section, we propose our UCB exploration strategy.

---

**Algorithm 1** Ensemble Voting

---
1: **Input**: $K \in \mathbb{N}_+$ copies of independently initialized $Q^*$-functions $\{Q_k\}_{k=1}^K$.
2: Let $B$ be a replay buffer storing transitions for training
3: **for** each episode **do** do
4:     Obtain initial state from environment $s_0$
5:     **for** step $t = 1, \dots$ until end of episode **do**
6:         Pick an action according to $a_t = \text{MajorityVote}(\{\text{argmax}_a\, Q_k(s_t, a)\}_{k=1}^K)$
7:         Execute $a_t$. Receive state $s_{t+1}$ and reward $r_t$ from the environment
8:         Add $(s_t, a_t, r_t, s_{t+1})$ to replay buffer $B$
9:         At learning interval, sample random minibatch and update $\{Q_k\}$
10:     **end for**
11: **end for**

---

## 4 UCB EXPLORATION STRATEGY USING $Q$-ENSEMBLES

In this section, we propose optimism-based exploration by adapting the UCB algorithms (Auer et al. (2002); Audibert et al. (2009)) from the bandit setting. The UCB algorithms maintain an upper-confidence bound for each arm, such that the expected reward from pulling each arm is smaller than this bound with high probability. At every time step, the agent optimistically chooses the arm with the highest UCB. Auer et al. (2002) constructed the UCB based on empirical reward and the number of times each arm is chosen. Audibert et al. (2009) incorporated the empirical variance of each arm's reward into the UCB, such that at time step $t$, an arm $A_t$ is pulled according to

$$A_t = \underset{i}{\text{argmax}} \left\{ \hat{r}_{i,t} + c_1 \cdot \sqrt{\frac{\hat{V}_{i,t} \log(t)}{n_{i,t}}} + c_2 \cdot \frac{\log(t)}{n_{i,t}} \right\}$$

where $\hat{r}_{i,t}$ and $\hat{V}_{i,t}$ are the empirical reward and variance of arm $i$ at time $t$, $n_{i,t}$ is the number of times arm $i$ has been pulled up to time $t$, and $c_1, c_2$ are positive constants.

We extend the intuition of UCB algorithms to the RL setting. Using the outputs of the $\{Q_k\}$ functions, we construct a UCB by adding the empirical standard deviation $\tilde{\sigma}(s_t, a)$ of $\{Q_k(s_t, a)\}_{k=1}^K$ to the empirical mean $\tilde{\mu}(s_t, a)$ of $\{Q_k(s_t, a)\}_{k=1}^K$. The agent chooses the action that maximizes this UCB

$$a_t \in \underset{a}{\text{argmax}} \left\{ \tilde{\mu}(s_t, a) + \lambda \cdot \tilde{\sigma}(s_t, a) \right\},$$

where $\lambda \in \mathbb{R}_+$ is a hyperparameter.

We present Algorithm 2, which incorporates the UCB exploration. The hyparparemeter $\lambda$ controls the degrees of exploration. In Section 5, we compare the performance of our algorithms on Atari games using a consistent set of parameters.

---

**Algorithm 2** UCB Exploration with $Q$-Ensembles

---
1: **Input:** Value function networks $Q$ with $K$ outputs $\{Q_k\}_{k=1}^K$. Hyperparameter $\lambda$.
2: Let $B$ be a replay buffer storing experience for training.
3: **for** each episode **do**
4:     Obtain initial state from environment $s_0$
5:     **for** step $t = 1, \dots$ until end of episode **do**
6:         Pick an action according to $a_t \in \text{argmax}_a \left\{ \tilde{\mu}(s_t, a) + \lambda \cdot \tilde{\sigma}(s_t, a) \right\}$
7:         Receive state $s_{t+1}$ and reward $r_t$ from environment, having taken action $a_t$
8:         Add $(s_t, a_t, r_t, s_{t+1})$ to replay buffer $B$
9:         At learning interval, sample random minibatch and update $\{Q_k\}$
10:     **end for**
11: **end for**

---

## 5 EXPERIMENT

In this section, we conduct experiments to answer the following questions:

1. does Ensemble Voting, Algorithm 1, improve upon existing algorithms including Double DQN and bootstrapped DQN?

2. is the proposed UCB exploration strategy of Algorithm 2 effective in improving learning compared to Algorithm 1, Double DQN and bootstrapped DQN?

3. how does UCB exploration compare with prior exploration methods such as the count-based exploration method of Bellemare et al. (2016)?

We evaluate the algorithms on each Atari game of the Arcade Learning Environment (Bellemare et al. (2013)). We use the multi-head neural net architecture of Osband et al. (2016). We fix the common hyperparameters of all algorithms based on a well-tuned double DQN implementation, which uses the Adam optimizer (Kingma and Ba (2014)), different learning rate and exploration schedules compared to Mnih et al. (2015). Appendix A tabulates the hyperparameters. The number of $\{Q_k\}$ functions is $K = 10$. Experiments are conducted on the OpenAI Gym platform (Brockman et al. (2016)) and trained with 40 million frames and 2 trials on each game.

We take the following directions to evaluate the performance of our algorithms:

1. we compare Algorithm 1 against Double DQN and bootstrapped DQN,

2. we isolate the impact of UCB exploration by comparing Algorithm 2 with $\lambda = 0.1$, denoted as `ucb exploration`, against Algorithm 1, Double DQN, and bootstrapped DQN.

3. we compare Algorithm 1 and Algorithm 2 with the count-based exploration method of Bellemare et al. (2016).

4. we aggregate the comparison according to different categories of games, to understand when our methods are suprior.

Figure 1 compares the normalized learning curves of all algorithms across Atari games. Overall, Ensemble Voting, Algorithm 1, outperforms both Double DQN and bootstrapped DQN. With exploration, `ucb exploration` improves further by outperforming Ensemble Voting.

In Appendix B, we tabulate detailed results that compare our algorithms, Ensemble Voting and `ucb exploration`, against prior methods. In Table 2, we tabulate the maximal mean reward in 100 consecutive episodes for Ensemble Voting, `ucb exploration`, bootstrapped DQN and Double DQN. Without exploration, Ensemble Voting already achieves higher maximal mean reward than both Double DQN and bootstrapped DQN in a majority of Atari games. Ensemble Voting performs better than Double DQN in 37 games out of the total 49 games evaluated, better than bootstrapped DQN in 41 games. `ucb exploration` achieves the highest maximal mean reward among these four algorithms in 30 games out of the total 49 games evaluated. Specifically, `ucb exploration` performs better than Double DQN in 38 out of 49 games evaluated, better than bootstrapped DQN in 45 games, and better than Ensemble Voting in 35 games. Figure 2 displays the learning curves of these five algorithms on a set of six Atari games. Ensemble Voting outperforms Double DQN and bootstrapped DQN. `ucb exploration` outperforms Ensemble Voting.

In Table 3, we compare our proposed methods with the count-based exploration method A3C+ of Bellemare et al. (2016) based on their published results of A3C+ trained with 200 million frames. We point out that even though our methods were trained with only 40 million frames, much less than A3C+'s 200 million frames, UCB exploration achieves the highest average reward in 28 games, Ensemble Voting in 10 games, and A3C+ in 10 games. Our approach outperforms A3C+.

Finally to understand why and when the proposed methods are superior, we aggregate the comparison results according to four categories: Human Optimal, Score Explicit, Dense Reward, and Sparse Reward. These categories follow the taxonomy in Table 1 of Ostrovski et al. (2017). Out of all games evaluated, 23 games are Human Optimal, 8 are Score Explicit, 8 are Dense Reward, and 5 are Sparse Reward. The comparison results are tabulated in Table 4, where we see `ucb exploration` achieves top performance in more games than Ensemble Voting, Double DQN, and Bootstrapped DQN in the categories of Human Optimal, Score Explicit, and Dense Reward. In Sparse Reward, both `ucb exploration` and Ensemble Voting achieve best performance in 2 games out of total of 5. Thus, we conclude that `ucb exploration` improves prior methods consistently across different game categories within the Arcade Learning Environment.

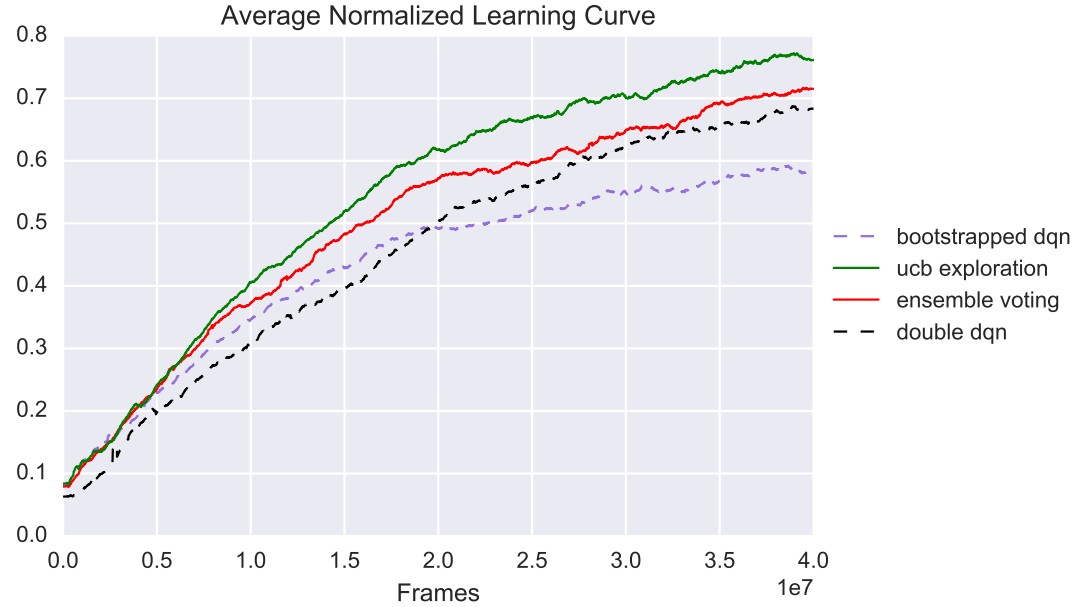

Figure 1: Comparison of algorithms in normalized learning curve. The normalized learning curve is calculated as follows: first, we normalize learning curves for all algorithms in the same game to the interval [0, 1]; next, average the normalized learning curve from all games for each algorithm.

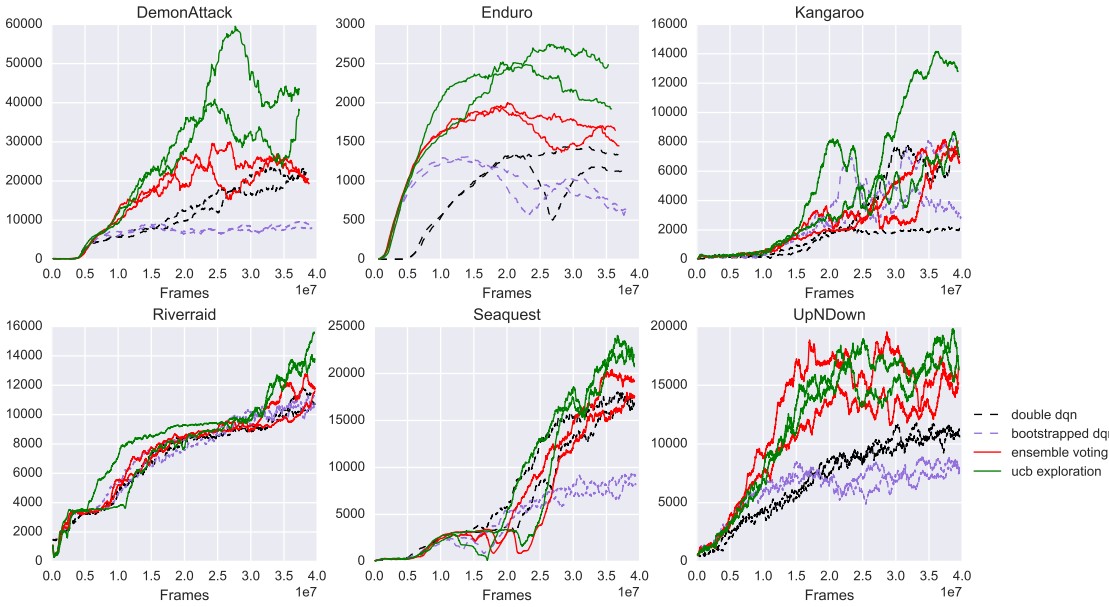

Figure 2: Comparison of UCB Exploration and Ensemble Voting against Double DQN and Bootstrapped DQN.

## 6  CONCLUSION

We proposed a *Q*-ensemble approach to deep *Q*-learning, a computationally practical algorithm inspired by Bayesian reinforcement learning that outperforms Double DQN and bootstrapped DQN, as evaluated on Atari. The key ingredient is the UCB exploration strategy, inspired by bandit algorithms. Our experiments show that the exploration strategy achieves improved learning performance on the majority of Atari games.

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

## A  HYPERPARAMETERS

We tabulate the hyperparameters in our well-tuned implementation of double DQN in Table 1:

| hyperparameter | value | descriptions |
|---|---|---|
| total training frames | 40 million | Length of training for each game. |
| minibatch size | 32 | Size of minibatch samples for each parameter update. |
| replay buffer size | 1000000 | The number of most recent frames stored in replay buffer. |
| agent history length | 4 | The number of most recent frames concatenated as input to the Q network. Total number of iterations = total training frames / agent history length. |
| target network update frequency | 10000 | The frequency of updating target network, in the number of parameter updates. |
| discount factor | 0.99 | Discount factor for Q value. |
| action repeat | 4 | Repeat each action selected by the agent this many times. A value of 4 means the agent sees every 4th frame. |
| update frequency | 4 | The number of actions between successive parameter updates. |
| optimizer | Adam | Optimizer for parameter updates. |
| $\beta_1$ | 0.9 | Adam optimizer parameter. |
| $\beta_2$ | 0.99 | Adam optimizer parameter. |
| $\epsilon$ | $10^{-4}$ | Adam optimizer parameter. |
| learning rate schedule | $\begin{cases} 10^{-4} & t \leq 10^6 \\ Interp(10^{-4}, 5*10^{-5}) & \text{otherwise} \\ 5*10^{-5} & t > 5*10^6 \end{cases}$ | Learning rate for Adam optimizer, as a function of iteration $t$. |
| exploration schedule | $\begin{cases} Interp(1, 0.1) & t < 10^6 \\ Interp(0.1, 0.01) & \text{otherwise} \\ 0.01 & t > 5*10^6 \end{cases}$ | Probability of random action in $\epsilon$-greedy exploration, as a function of the iteration $t$. |
| replay start size | 50000 | Number of uniform random actions taken before learning starts. |

Table 1: Double DQN hyperparameters. These hyperparameters are selected based on performances of seven Atari games: Beam Rider, Breakout, Pong, Enduro, Qbert, Seaquest, and Space Invaders. $Interp(\cdot, \cdot)$ is linear interpolation between two values.

# B  RESULTS TABLES

| | Bootstrapped DQN | Double DQN | Ensemble Voting | UCB-Exploration |
|---|---|---|---|---|
| Alien | 1445.1 | 2059.7 | 2282.8 | **2817.6** |
| Amidar | 430.58 | 667.5 | **683.72** | 663.8 |
| Assault | 2519.06 | 2820.61 | 3213.58 | **3702.76** |
| Asterix | 3829.0 | 7639.5 | **8740.0** | 8732.0 |
| Asteroids | 1009.5 | 1002.3 | **1149.3** | 1007.8 |
| Atlantis | 1314058.0 | 1982677.0 | 1786305.0 | **2016145.0** |
| Bank Heist | 795.1 | 789.9 | 869.4 | **906.9** |
| Battle Zone | 26230.0 | 24880.0 | **27430.0** | 26770.0 |
| Beam Rider | 8006.58 | 7743.74 | 7991.9 | **9188.26** |
| Bowling | 28.62 | 30.92 | 32.92 | **38.06** |
| Boxing | 85.91 | 94.07 | 94.47 | **98.08** |
| Breakout | 400.22 | **467.45** | 426.78 | 411.31 |
| Centipede | 5328.77 | 5177.51 | 6153.28 | **6237.18** |
| Chopper Command | 2153.0 | 3260.0 | 3544.0 | **3677.0** |
| Crazy Climber | 110926.0 | 124456.0 | 126677.0 | **127754.0** |
| Demon Attack | 9811.45 | 23562.55 | 30004.4 | **59861.9** |
| Double Dunk | -10.82 | -14.58 | -11.94 | **-4.08** |
| Enduro | 1314.31 | 1439.59 | 1999.88 | **2752.55** |
| Fishing Derby | 21.89 | 23.69 | **30.02** | 29.71 |
| Freeway | 33.57 | 32.93 | 33.92 | **33.96** |
| Frostbite | 1284.8 | 529.2 | 1196.0 | **1903.0** |
| Gopher | 7652.2 | 12030.0 | 10993.2 | **12910.8** |
| Gravitar | 227.5 | 279.5 | **371.5** | 318.0 |
| Ice Hockey | -4.62 | -4.63 | **-1.73** | -4.71 |
| Jamesbond | 594.5 | 594.0 | 602.0 | **710.0** |
| Kangaroo | 8186.0 | 7787.0 | 8174.0 | **14196.0** |
| Krull | 8537.52 | 8517.91 | 8669.17 | **9171.61** |
| Kung Fu Master | 24153.0 | **32896.0** | 30988.0 | 31291.0 |
| Montezuma Revenge | 2.0 | 4.0 | 1.0 | 4.0 |
| Ms Pacman | 2508.7 | 2498.1 | 3039.7 | **3425.4** |
| Name This Game | 8212.4 | **9806.9** | 9255.1 | 9570.5 |
| Pitfall | -5.99 | -7.57 | -3.37 | **-1.47** |
| Pong | 21.0 | 20.67 | 21.0 | 20.95 |
| Private Eye | 1815.19 | 788.63 | **1845.28** | 1252.01 |
| Qbert | 10557.25 | 6529.5 | 12036.5 | **14198.25** |
| Riverraid | 11528.0 | 11834.7 | 12785.8 | **15622.2** |
| Road Runner | 52489.0 | 49039.0 | **54768.0** | 53596.0 |
| Robotank | 21.03 | 29.8 | 31.83 | **41.04** |
| Seaquest | 9320.7 | 18056.4 | 20458.6 | **24001.6** |
| Space Invaders | 1549.9 | 1917.5 | 1890.8 | **2626.55** |
| Star Gunner | 20115.0 | **52283.0** | 41684.0 | 47367.0 |
| Tennis | -15.11 | -14.04 | -11.63 | **-7.8** |
| Time Pilot | 5088.0 | 5548.0 | 6153.0 | **6490.0** |
| Tutankham | 167.47 | **223.43** | 208.61 | 200.76 |
| Up N Down | 9049.1 | 11815.3 | 19528.3 | **19827.3** |
| Venture | **115.0** | 96.0 | 78.0 | 67.0 |
| Video Pinball | 364600.85 | **374686.89** | 343380.29 | 372564.11 |
| Wizard Of Wor | 2860.0 | 3877.0 | 5451.0 | **5873.0** |
| Zaxxon | 592.0 | **8903.0** | 3901.0 | 3695.0 |
| Times best | 1 | 7 | 9 | 30 |

Table 2: Comparison of maximal mean rewards achieved by agents. Maximal mean reward is calculated in a window of 100 consecutive episodes. Bold denotes the highest value in each row.

| | Ensemble Voting | UCB-Exploration | A3C+ |
|---|---|---|---|
| Alien | 2282.8 | **2817.6** | 1848.33 |
| Amidar | 683.72 | 663.8 | **964.77** |
| Assault | 3213.58 | **3702.76** | 2607.28 |
| Asterix | **8740.0** | 8732.0 | 7262.77 |
| Asteroids | 1149.3 | 1007.8 | **2257.92** |
| Atlantis | 1786305.0 | **2016145.0** | 1733528.71 |
| Bank Heist | 869.4 | 906.9 | **991.96** |
| Battle Zone | **27430.0** | 26770.0 | 7428.99 |
| Beam Rider | 7991.9 | **9188.26** | 5992.08 |
| Bowling | 32.92 | 38.06 | **68.72** |
| Boxing | 94.47 | **98.08** | 13.82 |
| Breakout | **426.78** | 411.31 | 323.21 |
| Centipede | 6153.28 | **6237.18** | 5338.24 |
| Chopper Command | 3544.0 | 3677.0 | **5388.22** |
| Crazy Climber | 126677.0 | **127754.0** | 104083.51 |
| Demon Attack | 30004.4 | **59861.9** | 19589.95 |
| Double Dunk | -11.94 | **-4.08** | -8.88 |
| Enduro | 1999.88 | **2752.55** | 749.11 |
| Fishing Derby | **30.02** | 29.71 | 29.46 |
| Freeway | 33.92 | **33.96** | 27.33 |
| Frostbite | 1196.0 | **1903.0** | 506.61 |
| Gopher | 10993.2 | **12910.8** | 5948.40 |
| Gravitar | **371.5** | 318.0 | 246.02 |
| Ice Hockey | **-1.73** | -4.71 | -7.05 |
| Jamesbond | 602.0 | 710.0 | **1024.16** |
| Kangaroo | 8174.0 | **14196.0** | 5475.73 |
| Krull | 8669.17 | **9171.61** | 7587.58 |
| Kung Fu Master | 30988.0 | **31291.0** | 26593.67 |
| Montezuma Revenge | 1.0 | 4.0 | **142.50** |
| Ms Pacman | 3039.7 | **3425.4** | 2380.58 |
| Name This Game | 9255.1 | **9570.5** | 6427.51 |
| Pitfall | -3.37 | **-1.47** | -155.97 |
| Pong | **21.0** | 20.95 | 17.33 |
| Private Eye | **1845.28** | 1252.01 | 100.0 |
| Qbert | 12036.5 | 14198.25 | **15804.72** |
| Riverraid | 12785.8 | **15622.2** | 10331.56 |
| Road Runner | **54768.0** | 53596.0 | 49029.74 |
| Robotank | 31.83 | **41.04** | 6.68 |
| Seaquest | 20458.6 | **24001.6** | 2274.06 |
| Space Invaders | 1890.8 | **2626.55** | 1466.01 |
| Star Gunner | 41684.0 | 47367.0 | **52466.84** |
| Tennis | -11.63 | **-7.8** | -20.49 |
| Time Pilot | 6153.0 | **6490.0** | 3816.38 |
| Tutankham | **208.61** | 200.76 | 132.67 |
| Up N Down | 19528.3 | **19827.3** | 8705.64 |
| Venture | **78.0** | 67.0 | 0.00 |
| Video Pinball | 343380.29 | **372564.11** | 35515.92 |
| Wizard Of Wor | 5451.0 | **5873.0** | 3657.65 |
| Zaxxon | 3901.0 | 3695.0 | **7956.05** |
| Times Best | 10 | 28 | 10 |

Table 3: Comparison of Ensemble Voting, UCB Exploration, both trained with 40 million frames and A3C+ of Bellemare et al. (2016), trained with 200 million frames

| Category | Total | Bootstrapped DQN | Double DQN | Ensemble Voting | UCB-Exploration |
|---|---|---|---|---|---|
| Human Optimal | 23 | 0 | 3 | 5 | 15 |
| Score Explicit | 8 | 0 | 2 | 1 | 5 |
| Dense Reward | 8 | 0 | 1 | 1 | 6 |
| Sparse Reward | 5 | 1 | 0 | 2 | 2 |

Table 4: Comparison of each method across different game categories. The Atari games are separated into four categories: human optimal, score explicit, dense reward, and sparse reward. In each row, we present the number of games in this category, the total number of games where each algorithm achieves the optimal performance according to Table 2. The game categories follow the taxonomy in Table 1 of Ostrovski et al. (2017)

## C   APPROXIMATING BAYESIAN $Q$-LEARNING WITH $Q$-ENSEMBLES

In this section, we first derive a posterior update formula for the $Q^*$-function under full exploration assumption and this formula turns out to depend on the transition Markov chain. Next, we approximate the posterior update with $Q$-ensembles $\{Q_k\}$ and demonstrate that the Bellman equation emerges as the approximate update rule for each $Q_k$.

### C.1   POSTERIOR UPDATE FOR THE $Q^*$-FUNCTION

An MDP is specified by the transition probability $T$ and the reward function $R$. Unlike prior works outlined in Section 2.3 which learned the posterior of the MDP, we will consider the joint distribution over $(Q^*, T)$. Note that $R$ can be recovered from $Q^*$ given $T$. So $(Q^*, T)$ determines a unique MDP. In this section, we assume that the agent samples $(s, a)$ according to a fixed distribution. The corresponding reward $r$ and next state $s'$ given by the MDP append to $(s, a)$ to form a transition $\tau = (s, a, r, s')$, for updating the posterior of $(Q^*, T)$. Recall that the $Q^*$-function satisfies the Bellman equation

$$Q(s, a) = r + \mathbb{E}_{s' \sim T(\cdot|s,a)} \left[ \gamma \max_{a'} Q(s', a') \right].$$

Denote the joint prior distribution as $p(Q^*, T)$ and the posterior as $\tilde{p}$. We apply Bayes' formula to expand the posterior:

$$\tilde{p}(Q^*, T|\tau) = \frac{p(\tau|Q^*, T) \cdot p(Q^*, T)}{Z(\tau)}$$
$$= \frac{p(Q^*, T) \cdot p(s'|Q^*, T, (s,a)) \cdot p(r|Q^*, T, (s,a,s')) \cdot p(s,a)}{Z(\tau)}, \quad (1)$$

where $Z(\tau)$ is a normalizing constant and the second equality is because $s$ and $a$ are sampled randomly from $\mathcal{S}$ and $\mathcal{A}$. Next, we calculate the two conditional probabilities in (1). First,

$$p(s'|Q^*, T, (s,a)) = p(s'|T, (s,a)) = T(s'|s,a), \quad (2)$$

where the first equality is because given $T$, $Q^*$ does not influence the transition. Second,

$$p(r|Q^*, T, (s,a,s')) = p(r|Q^*, T, (s,a))$$
$$= \mathbb{1}_{\{Q^*(s,a)=r+\gamma \cdot \mathbb{E}_{s'' \sim T(\cdot|s,a)} \max_{a'} Q^*(s'',a')\}}$$
$$:= \mathbb{1}(Q^*, T), \quad (3)$$

where $\mathbb{1}_{\{\cdot\}}$ is the indicator function and in the last equation we abbreviate it as $\mathbb{1}(Q^*, T)$. Substituting (2) and (3) into (1), we obtain the joint posterior of $Q^*$ and $T$ after observing an additional randomly sampled transition $\tau$

$$\tilde{p}(Q^*, T|\tau) = \frac{p(Q^*, T) \cdot T(s'|s,a) \cdot p(s,a)}{Z(\tau)} \cdot \mathbb{1}(Q^*, T). \quad (4)$$

### C.2   APPROXIMATIONS WITH $Q$-ENSEMBLES

The exact $Q^*$-posterior update (4) is intractable in high-dimensional RL due to the large space of $(Q^*, T)$. Thus, we make several approximations to the $Q^*$-posterior update. First, we approximate

the prior of $Q^*$ by sampling $K \in \mathbb{N}_+$ independently initialized $Q^*$-functions $\{Q_k\}_{k=1}^K$. Next, we update them as more transitions are sampled. The resulting $\{Q_k\}$ approximate samples drawn from the posterior. The agent chooses the action by taking a majority vote from the actions determined by each $Q_k$.

We derive the update rule for $\{Q_k\}$ after observing a new transition $\tau = (s, a, r, s')$. At iteration $i$, given $Q^* = Q_{k,i}(\cdot; \theta_k)$ parametrized by $\theta_k$ the joint probability of $(Q^*, T)$ factors into

$$p(Q_{k,i}, T) = p(Q^*, T | Q^* = Q_{k,i}) = p(T | Q_{k,i}). \tag{5}$$

Substitute (5) into (4) and we obtain the corresponding posterior for each $Q_{k,i+1}$ at iteration $i + 1$ as

$$\tilde{p}(Q_{k,i+1}, T | \tau) = \frac{p(T | Q_{k,i}) \cdot T(s' | s, a) \cdot p(s, a)}{Z(\tau)} \cdot \mathbb{1}(Q_{k,i+1}, T). \tag{6}$$

$$\tilde{p}(Q_{k,i+1} | \tau) = \int_T \tilde{p}(Q_{k,i+1}, T | \tau) \mathrm{d}T = p(s, a) \cdot \int_T \tilde{p}(T | Q_{k,i}, \tau) \cdot \mathbb{1}(Q_{k,i+1}, T) \mathrm{d}T. \tag{7}$$

We update $Q_{k,i}$ to $Q_{k,i+1}$ according to

$$Q_{k,i+1} \leftarrow \underset{Q_{k,i+1}}{\operatorname{argmax}} \, \tilde{p}(Q_{k,i+1} | \tau). \tag{8}$$

We first derive a lower bound of the the posterior $\tilde{p}(Q_{k,i+1} | \tau)$:

$$\tilde{p}(Q_{k,i+1} | \tau) = p(s, a) \cdot \mathbb{E}_{T \sim \tilde{p}(T | Q_{k,i}, \tau)} \, \mathbb{1}(Q_{k,i+1}, T)$$

$$= p(s, a) \cdot \mathbb{E}_{T \sim \tilde{p}(T | Q_{k,i}, \tau)} \lim_{c \to +\infty} \exp\left( -c[Q_{k,i+1}(s, a) - r - \gamma \, \mathbb{E}_{s'' \sim T(\cdot | s, a)} \max_{a'} Q_{k,i+1}(s'', a')]^2 \right)$$

$$= p(s, a) \cdot \lim_{c \to +\infty} \mathbb{E}_{T \sim \tilde{p}(T | Q_{k,i}, \tau)} \exp\left( -c[Q_{k,i+1}(s, a) - r - \gamma \, \mathbb{E}_{s'' \sim T(\cdot | s, a)} \max_{a'} Q_{k,i+1}(s'', a')]^2 \right)$$

$$\geq p(s, a) \cdot \lim_{c \to +\infty} \exp\left( -c \, \mathbb{E}_{T \sim \tilde{p}(T | Q_{k,i}, \tau)} [Q_{k,i+1}(s, a) - r - \gamma \, \mathbb{E}_{s'' \sim T(\cdot | s, a)} \max_{a'} Q_{k,i+1}(s'', a')]^2 \right)$$

$$= p(s, a) \cdot \mathbb{1}_{\mathbb{E}_{T \sim \tilde{p}(T | Q_{k,i}, \tau)} [Q_{k,i+1}(s,a) - r - \gamma \, \mathbb{E}_{s'' \sim T(\cdot | s, a)} \max_{a'} Q_{k,i+1}(s'', a')]^2 = 0}. \tag{9}$$

where we apply a limit representation of the indicator function in the third equation. The fourth equation is due to the bounded convergence theorem. The inequality is Jensen's inequality. The last equation (9) replaces the limit with an indicator function.

A sufficient condition for (8) is to maximize the lower-bound of the posterior distribution in (9) by ensuring the indicator function in (9) to hold. We can replace (8) with the following update

$$Q_{k,i+1} \leftarrow \underset{Q_{k,i+1}}{\operatorname{argmin}} \, \mathbb{E}_{T \sim \tilde{p}(T | Q_{k,i}, \tau)} \left[ Q_{k,i+1}(s, a) - \left( r + \gamma \cdot \mathbb{E}_{s'' \sim T(\cdot | s, a)} \max_{a'} Q_{k,i+1}(s'', a') \right) \right]^2. \tag{10}$$

However, (10) is not tractable because the expectation in (10) is taken with respect to the posterior $\tilde{p}(T | Q_{k,i}, \tau)$ of the transition $T$. To overcome this challenge, we approximate the posterior update by reusing the one-sample next state $s'$ from $\tau$. Solving the exact minimal for each $Q_{k,i+1}$ is impractical, thus we take a gradient step on $Q_{k,i+1}$ according to the following gradient

$$\theta_k \leftarrow \theta_k + \eta \cdot \left( Q_k(s, a; \theta_k) - (r + \gamma \cdot \max_{a'} Q_k(s', a'; \theta_k)) \right) \nabla_{\theta_k} Q_k(s, a; \theta_k),$$

where $\eta$ is the step size. Instead of updating $Q_k$ after each transition, we use an experience replay buffer $B$ to store observed transitions and sample a minibatch $B_{\text{mini}}$ of transitions $(s, a, r, s')$ for each update. In this case, the batched update of each $Q_{k,i}$ to $Q_{k,i+1}$ becomes a standard Bellman update

$$\theta_k \leftarrow \theta_k + \eta \cdot \mathbb{E}_{(s,a,r,s') \in B_{\text{mini}}} \left[ \left( Q_k(s, a; \theta_k) - (r + \gamma \cdot \max_{a'} Q_k(s', a'; \theta_k)) \right) \nabla_{\theta_k} Q_k(s, a; \theta_k) \right].$$

## D    INFOGAIN EXPLORATION

In this section, we also studied an "InfoGain" exploration bonus, which encourages agents to gain information about the $Q^*$-function and examine its effectiveness. We found it had some benefits on top of Ensemble Voting, but no uniform additional benefits once already using Q-ensembles on top of Double DQN. We describe the approach and our experimental findings.

Similar to Sun et al. (2011), we define the information gain from observing an additional transition $\tau_n$ as

$$H_{\tau_t|\tau_1,\ldots,\tau_{n-1}} = D_{KL}(\tilde{p}(Q^*|\tau_1,\ldots,\tau_n)||\tilde{p}(Q^*|\tau_1,\ldots,\tau_{n-1}))$$

where $\tilde{p}(Q^*|\tau_1,\ldots,\tau_n)$ is the posterior distribution of $Q^*$ after observing a sequence of transitions $(\tau_1,\ldots,\tau_n)$. The total information gain is

$$H_{\tau_1,\ldots,\tau_N} = \sum_{n=1}^{N} H_{\tau_n|\tau_1,\ldots,\tau_{n-1}}. \tag{11}$$

Our Ensemble Voting, Algorithm 1, does not maintain the posterior $\tilde{p}$, thus we cannot calculate (11) explicitly. Instead, inspired by Lakshminarayanan et al. (2016), we define an InfoGain exploration bonus that measures the disagreement among $\{Q_k\}$. Note that

$$H_{\tau_1,\ldots,\tau_N} + \mathsf{H}(\tilde{p}(Q^*|\tau_1,\ldots,\tau_N)) = \mathsf{H}(p(Q^*)),$$

where $\mathsf{H}(\cdot)$ is the entropy. If $H_{\tau_1,\ldots,\tau_N}$ is small, then the posterior distribution has high entropy and high residual information. Since $\{Q_k\}$ are approximate samples from the posterior, high entropy of the posterior leads to large discrepancy among $\{Q_k\}$. Thus, the exploration bonus is monotonous with respect to the residual information in the posterior $\mathsf{H}(\tilde{p}(Q^*|\tau_1,\ldots,\tau_N))$. We first compute the Boltzmann distribution for each $Q_k$

$$P_{\mathsf{T},k}(a|s) = \frac{\exp\left(Q_k(s,a)/\mathsf{T}\right)}{\sum_{a'}\exp\left(Q_k(s,a')/\mathsf{T}\right)},$$

where $\mathsf{T} > 0$ is a temperature parameter. Next, calculate the average Boltzmann distribution

$$P_{\mathsf{T},\mathrm{avg}} = \frac{1}{K}\cdot\sum_{k=1}^{K} P_{\mathsf{T},k}(a|s).$$

The InfoGain exploration bonus is the average KL-divergence from $\{P_{\mathsf{T},k}\}_{k=1}^{K}$ to $P_{\mathsf{T},\mathrm{avg}}$

$$b_{\mathsf{T}}(s) = \frac{1}{K}\cdot\sum_{k=1}^{K} \mathrm{D}_{KL}[P_{\mathsf{T},k}||P_{\mathsf{T},\mathrm{avg}}]. \tag{12}$$

The modified reward is

$$\hat{r}(s,a,s') = r(s,a) + \rho\cdot b_{\mathsf{T}}(s), \tag{13}$$

where $\rho \in \mathbb{R}_+$ is a hyperparameter that controls the degree of exploration.

The exploration bonus $b_{\mathsf{T}}(s_t)$ encourages the agent to explore where $\{Q_k\}$ disagree. The temperature parameter $\mathsf{T}$ controls the sensitivity to discrepancies among $\{Q_k\}$. When $\mathsf{T} \to +\infty$, $\{P_{\mathsf{T},k}\}$ converge to the uniform distribution on the action space and $b_{\mathsf{T}}(s) \to 0$. When $\mathsf{T}$ is small, the differences among $\{Q_k\}$ are magnified and $b_{\mathsf{T}}(s)$ is large.

We display Algorithrim 3, which incorporates our InfoGain exploration bonus into Algorithm 2. The hyperparameters $\lambda$, $\mathsf{T}$ and $\rho$ vary for each game.

---

**Algorithm 3** UCB + InfoGain Exploration with $Q$-Ensembles

---

1: **Input:** Value function networks $Q$ with $K$ outputs $\{Q_k\}_{k=1}^{K}$. Hyperparameters $\mathsf{T}, \lambda$, and $\rho$.
2: Let $B$ be a replay buffer storing experience for training.
3: **for** each episode **do**
4:      Obtain initial state from environment $s_0$
5:      **for** step $t = 1,\ldots$ until end of episode **do**
6:          Pick an action according to $a_t \in \mathrm{argmax}_a\left\{\tilde{\mu}(s_t,a) + \lambda\cdot\tilde{\sigma}(s_t,a)\right\}$
7:          Receive state $s_{t+1}$ and reward $r_t$ from environment, having taken action $a_t$
8:          Calculate exploration bonus $b_{\mathsf{T}}(s_t)$ according to (12)
9:          Add $(s_t, a_t, r_t + \rho\cdot b_{\mathsf{T}}(s_t), s_{t+1})$ to replay buffer $B$
10:         At learning interval, sample random minibatch and update $\{Q_k\}$
11:      **end for**
12: **end for**

---

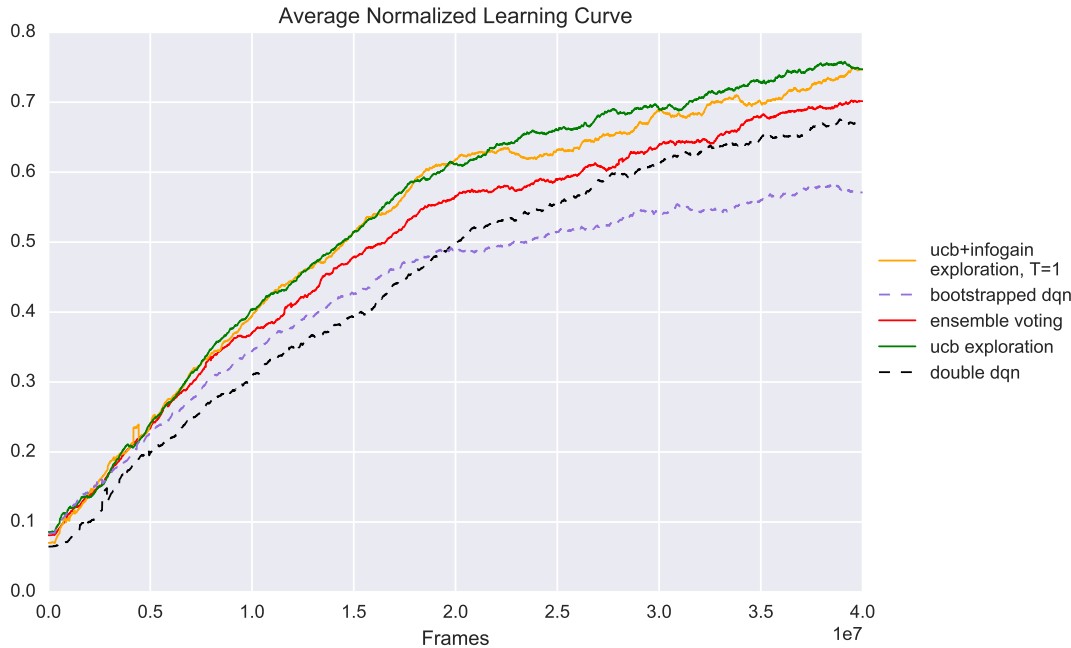

Figure 3: Comparison of all algorithms in normalized curve. The normalized learning curve is calculated as follows: first, we normalize learning curves for all algorithms in the same game to the interval $[0, 1]$; next, average the normalized learning curve from all games for each algorithm.

## D.1 PERFORMANCE OF UCB+INFOGAIN EXPLORATION

We demonstrate the performance of the combined UCB+InfoGain exploration in Figure 3 and Figure 3. We augment the previous figures in Section 5 with the performance of `ucb+infogain exploration`, where we set $\lambda = 0.1, \rho = 1$, and $\mathsf{T} = 1$ in Algorithm 3.

Figure 3 shows that combining UCB and InfoGain exploration does not lead to uniform improvement in the normalized learning curve.

At the individual game level, Figure 3 shows that the impact of InfoGain exploration varies. UCB exploration achieves sufficient exploration in games including Demon Attack and Kangaroo and Riverraid, while InfoGain exploration further improves learning on Enduro, Seaquest, and Up N Down. The effect of InfoGain exploration depends on the choice of the temperature $\mathsf{T}$. The optimal temperature parameter varies across games. In Figure 5, we display the behavior of `ucb+infogain exploration` with different temperature values. Thus, we see the InfoGain exploration bonus, tuned with the appropriate temperature parameter, can lead to improved learning for games that require extra exploration, such as ChopperCommand, KungFuMaster, Seaquest, UpNDown.

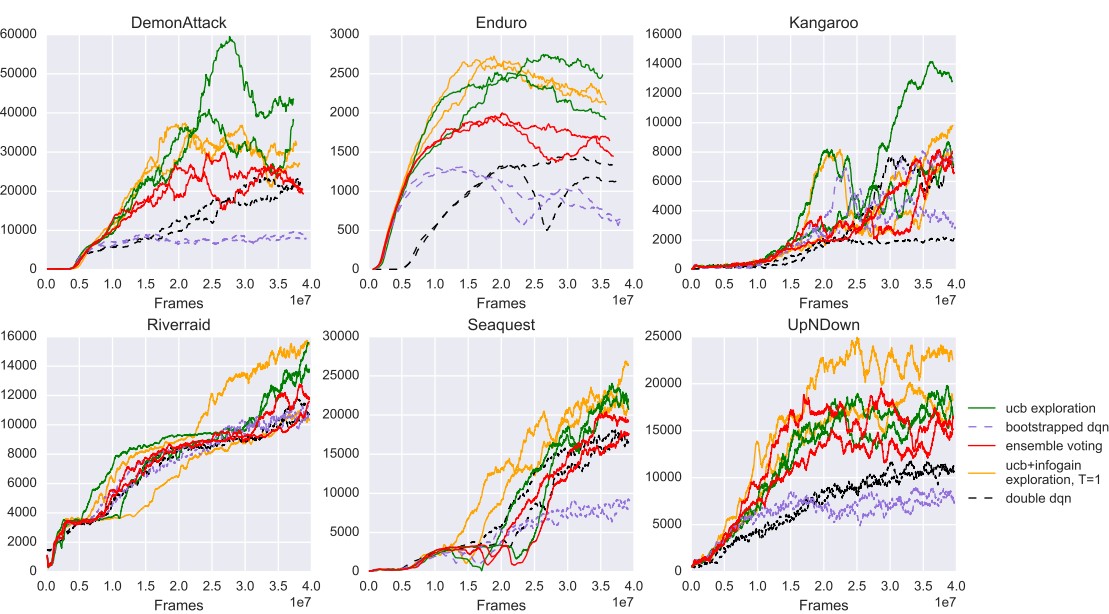

Figure 4: Comparison of algorithms against Double DQN and bootstrapped DQN.

## D.2 UCB+INFOGAIN EXPLORATION WITH DIFFERENT TEMPERATURES

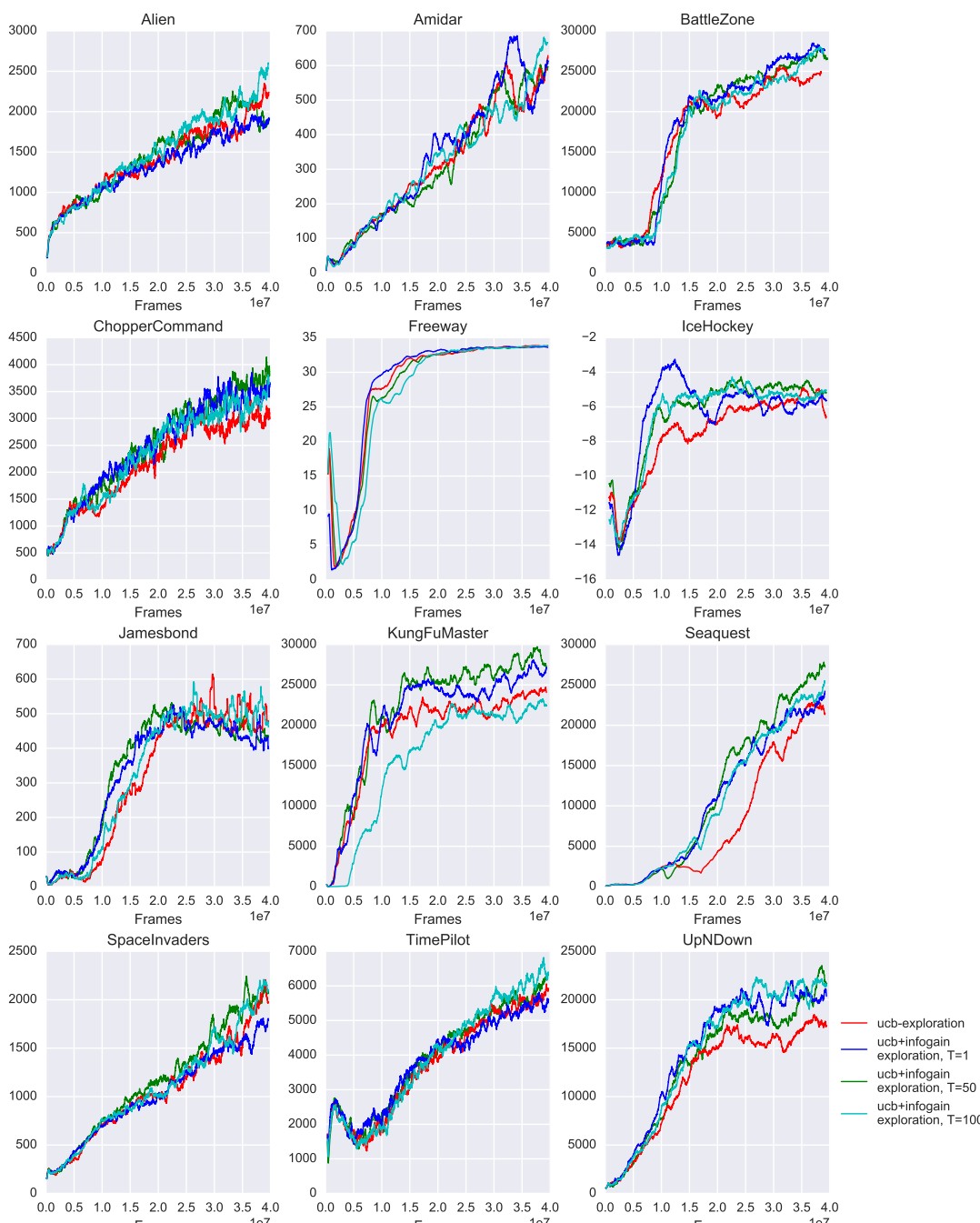

Figure 5: Comparison of UCB+InfoGain exploration with different temperatures versus UCB exploration.

