# OpenReview forum: "UCB EXPLORATION VIA Q-ENSEMBLES"
_ICLR.cc/2018/Conference — Reject_

### Official Review · AnonReviewer2 · 2017-11-27
**A good paper that shows UCB-ensemble outperforms Thompson Sampling ensemble (like bootstrapped DQN) in experiments.**

**Rating:** 6
**Confidence:** 5

**Review:**

This paper paper uses an ensemble of networks to represent the uncertainty in deep reinforcement learning.
The algorithm then chooses optimistically over the distribution induced by the ensemble.
This leads to improved learning / exploration, notably better than the similar approach bootstrapped DQN.

There are several things to like about this paper:
- It is a clear paper, with a simple message and experiments that back up the claims.
- The proposed algorithm is simple and could be practical in a lot of settings and even non-DQN variants.
- It is interesting that Bootstrapped DQN gets such poor performance, this suggests that it is very important in the original paper https://arxiv.org/abs/1602.04621 that "ensemble voting" is applied to the test evaluation... (why do you think this is by the way, do you think it has something to do with the data being *more* off-policy / diverse under a TS vs UCB scheme?)

On the other hand:
- The novelty/scope of this work is somewhat limited... this is more likely (valuable) incremental work than a game-changer.
- Something feels wrong/hacky/incomplete about just doing "ensemble" for uncertainty without bootstrapping/randomization... if we had access to more powerful optimization techniques then this certainly wouldn't be sensible - I think that you should mention that you are heavily reliant on "random initialization + SGD/Adam + specific network architecture" to maintain this idea of uncertainty. For example, this wouldn't work for linear value functions!
- I think the original bootstrapped DQN used "ensemble voting" at test time, so maybe you should change the labels or the way this is introduced/discussed. It's definitely very interesting that *essentially* the learning benefit is coming from ensembling (rather than "raw" bootstrapped DQN) and UCB still looks like it does better.
- I'm not convinced that page 4 and the "Bayesian" derivation really add too much value to this paper... alternatively, maybe you could introduce the actual algorithm first (train K models in parallel) and then say "this is similar to particle filter" and add the mathematical derivation after, rather than as if it was some complex formula derived. If you want to reference some justification/theory for ensemble-based uncertainty approximation you might consider https://arxiv.org/pdf/1705.07347.pdf instead.
- I think this paper might miss the point of the "bigger" problem of efficient exploration in RL... or even how to get "deep" exploration with deep RL. Yes this algorithm sees improvements across Atari, but it's not clear why/if this is a step change versus simply increasing the amount of replay or tuning the learning rate.  (Actually I do believe this algorithm can demonstrate deep exploration... but it looks like we're not seeing the big improvements on the "sub-human" games you might hope.)

Overall I do think this is a pretty good short paper/evaluation of UCB-ensembles on Atari.
The scope/insight of the paper isn't groundbreaking, but I think it delivers a clear short message on the Atari benchmark.
Perhaps this will encourage people to dig deeper into some of these issues... I vote accept.

---

> ### Author Response · Authors · 2018-01-04
> **Reply to the review**
>
> We thank the reviewer’s comments and address in the following:
>
> 1. Bootstrapped DQN samples one Q network from the ensemble applies it for a whole episode for exploration. We hypothesize that this intuition of deep exploration, by consistently using one Q function in each episode, does not guarantee that each Q function’s exploration is beneficial nor efficient. For example, each Q function deviates from the ensembled Q and accumulates inefficiency in a long episode. Although our proposed methods use the same network structure of bootstrapped DQN, the goal is very different: we build exploration bonus based on the uncertainty or discrepancy of Q ensembles. In UCB exploration, exploration bonus is based on the uncertainty of Q ensembles and encourages the agent to reduce the uncertainty in Q values.
>
> 2. Ensemble uncertainty is due to Q networks being parametrized with deep neural networks, which introduces nonconvexity in Bellman update. Thus, even though the Q networks are trained with the same samples, their parameters do not converge to the same. We also experimented with training each Q network with independently sampled transitions from the reply buffer, and did not observe improved performance. We don’t think the optimization method (SGD/Adam) plays a key role. This phenomenon that bagging worsens the performance of deep ensembles is also observed in supervised training setting. [Lee et al, 2015] observed that supervised learning trained with deep ensembles with random initializations perform better than bagging for deep ensembles. [Balaji et al, 2017] used deep ensembles for uncertainty estimates and also observed that bagging deteriorated performance in their experiments. We will revise and clarify the source of uncertainty from the ensembles.
>
> 3. We will modify/shorten the derivation on pages 3 and 4.
>
> 4. On efficient exploration in RL, our proposed two algorithms use the Q functions directly while prior works construct exploration bonus using state-visitation counts, which are not tied to the rewards that agents seek to maximize. Our goal is to construct methods that reduce the inefficiency of prior algorithms where learning can be wasted on visiting irrelevant states. Thus, by improving upon bootstrapped DQN and comparing with state-visitation count-based methods such as A3C+, we demonstrate that this direction of exploration based on Q-values is promising, and different from hyperparameter tuning. Due to compute constraint, we trained the proposed algorithms on each game with 40 million frames, less than 200 million frames used in prior works. Thus games that typically require more frames to learn do not show big improvement in our experiments.
>
> References:
> S. Lee, S. Purushwalkam, M. Cogswell, D. Crandall, and D. Batra. Why M heads are better than one: Training a diverse ensemble of deep networks. arXiv preprint arXiv:1511.06314, 2015.
>
> Lakshminarayanan, Balaji, Alexander Pritzel, and Charles Blundell. "Simple and scalable predictive uncertainty estimation using deep ensembles." Advances in Neural Information Processing Systems. 2017.

---

### Official Review · AnonReviewer1 · 2017-11-27
**New exploration method for DeepRL with some good results.**

**Rating:** 7
**Confidence:** 4

**Review:**

The authors propose a new exploration algorithm for Deep RL. They maintain an ensemble of Q-values (based on different initialisations) to model uncertainty over Q. The ensemble is then used to derive a confidence interval at each step, which is used to select actions UCB-style.

There is some attempt at a Bayesian interpretation for the Bellman update. But to me it feels a bit like shoehorning the probabilistic interpretation into an already existing update - I’m not sure this is justified and necessary here. Moreover, the UCB strategy is generally not considered a Bayesian strategy, so I wasn’t convinced by the link to Bayesian RL in this paper.

I liked the actual proposed method otherwise, and the experimental results on Atari seem good (but see also latest SOTA Atari results, for example the Rainbow paper). Some questions about the results:
-How does it perform compared to epsilon-greedy added on top of Alg1, or is there evidence that this produces any meaningful exploration versus noise?
-How does the distribution of Q values look like during different phases of learning?
-Was epsilon-greedy used in addition to UCB exploration? Question for both Alg 1 and Alg 2.
-What’s different between Alg 1 and bootstrapped DQN (other than the action selection)?

Minor things:
-Missing propto in Eq 7?
-Maybe mention that the leftarrows are not hard updates. Maybe you already do somewhere…
-it looks more a Bellman residual update as written in (11).

---

> ### Public Comment · (anonymous) · 2017-12-19
> **UCB**
>
> Regarding your comment on
> "There is some attempt at a Bayesian interpretation for the Bellman update. But to me it feels a bit like shoehorning the probabilistic interpretation into an already existing update - I’m not sure this is justified and necessary here. Moreover, the UCB strategy is generally considered a Bayesian strategy, so I wasn’t convinced by the link to Bayesian RL in this paper."
>
> As you mentioned, it is not clear to me as well what is the purpose if the derivations on pages 3 and 4 where it ends up to equation (12). But regarding your latter statement, could you please point me to a reference which says UCB strategy is a Bayesian strategy?
>
> Cheers

---

> > ### Comment · AnonReviewer1 · 2017-12-19
> > **RE: UCB**
> >
> > Sorry for the confusion, I meant *not* considered a Bayesian strategy of course... I've edited my review.

---

> ### Author Response · Authors · 2018-01-04
> **Reply to the review**
>
> We thank the reviewer’s comment and address in the following:
>
> 1. We will modify/shorten the derivation on pages 3 and 4.
>
> 2. We observed the Q values for actions chosen according to Alg 1 and Alg 2. These Q values correspond to good actions. During learning, the Q values for such good actions gradually increase. The discrepancies between the Q values also increase in absolute values. But normalized by the mean Q value from different Q networks, the discrepancies gradually decrease.
>
> 3. In Alg 1 and Alg 2, epsilon-greedy is not used, such that we can isolate the effects of exploration using Ensemble Voting or UCB exploration only. We did not experiment with adding epsilon-greedy on top of Alg 1, but agree that it will be an interesting experiment to see whether epsilon-greedy helps or hurts exploration on top of Alg 1.
>
> 4. Besides action selection, bootstrapped DQN allows each Q network to be trained with different samples (using a masking mechanism), even though in bootstrapped DQN’s Atari experiments, all Q networks are trained using the same samples. In Alg 1, Q networks only use random initialization and trained with the same samples. We also experimented with training Q networks with independently drawn samples, which deteriorated the performance.

---

### Official Review · AnonReviewer3 · 2017-12-01
**This paper shows improvement over baselines. But does not seem to offer significant insight or dramatic improvement.**

**Rating:** 5
**Confidence:** 3

**Review:**

This paper introduces a number of different techniques for improving exploration in deep Q learning. The main technique is to use UCB (upper confidence bound) to speedup exploration. The authors also introduces "Ensemble voting" facilitate exploitation.

This paper shows improvement over baselines. But does not seem to offer significant insight or dramatic improvement. The techniques introduced are a small permutation of previous results. The baselines are not particularly strong either.

The paper appeared to have be rushed. The presentation is not always clear.

I also have the following questions I hope the authors could help me with:

1. I failed to understand how Eqn (5). Could you please clarify.

2. What is the significance of the math introduced in section 3? All that was proposed was: (1) Majority voting, (2) UCB exploration.

3. Why comparing to A3C+ which is not necessarily better than A3C in final performance?

4. Why not comparing to Bootstrapped DQN since the proposed method is based on it?

5. Why is the proposed method better than Bootstrapped DQN, since UCB does not necessarily outperform Thompson sampling in the case of bandits?

6. If there is a section on INFOGAIN exploration, why not mention it in the main text?

---

> ### Author Response · Authors · 2018-01-04
> **Reply to the review**
>
> We thank the reviewer’s comments. We address in the following:
>
> 1. We first comment that the improvement from our proposed methods is significant. We used a strong Double DQN baseline, which achieves competitive or better learning results trained with 40 million frames, compared with prior published results [Van Hasselt, et al, 2016] trained with 200 million frames.  Improvement of proposed methods is significant over this strong Double DQN baseline. Table 2 in Appendix B shows that Ensemble Voting performs better than Double DQN in 37 out of 49 games evaluated, and UCB Exploration performs better than Double DQN in 38 out of 49 games evaluated. In addition, UCB Exploration performs better than Ensemble Voting in 35 out of 49 games evaluated. We will include such comparison in the results section.
>
> 2. We also compared to bootstrapped DQN as shown in Figure 1, Figure 2, and the results Table 2 in Appendix B.
>
> 3. A3C+ represents one line of research comprised of multiple works where the agent constructs exploration bonus based on state visitation counts. As discussed in Section 2.2, the exploration bonus from these methods does not depend on the reward, thus the exploration may focus on irrelevant aspects of the environment. In comparison, our exploration bonus depend on the Q values directly. We chose A3C+ to compare our method of reward-based exploration bonus against count-based exploration bonus and demonstrate that this reward/Q values-based approach of constructing exploration bonus is promising.
>
> 4. Bootstrapped DQN samples one Q network from the ensemble applies it for a whole episode for exploration. We hypothesize that this intuition of deep exploration, by consistently using one Q function in each episode, does not guarantee that exploration is beneficial nor efficient. For example, each Q function deviates from the ensembled-Q. Although our proposed methods use the same network structure of bootstrapped DQN, the goal is very different: we build exploration bonus based on the uncertainty or discrepancy of Q ensembles. UCB exploration bonus is based on the uncertainty of Q ensembles and encourages the agent to reduce the uncertainty in Q values.
>
> 5. The INFOGAIN section attempts another approach of exploration using Q-ensembles. However, the improvement of this method is less consistently across the board. This could be due to the approximations we made in constructing the INFOGAIN exploration bonus. We document the results of the experiment in the Appendix for potential future interest in this direction.
>
>
> 6. We will modify/shorten the derivation on pages 3 and 4.

---

### Public Comment · (anonymous) · 2017-12-19
**Few clarifications**

In the main text, the authors mentioned that
"A sufficient condition for (8) is to maximize the lower-bound of the posterior distribution in (9) by ensuring the indicator function in (9) to hold."
The input to the indicator function looks like the second moment of a random variable. Could you elaborate when it happens? I am not sure it can be achieved for stochastic reward. Is that correct?
Furthermore, the paper suggests that "We can replace (8) with the update (10)". Can you comment on why this is the case? Is it again for the deterministic reward? In Alg2, line 9, when you update according (12), you mean using TD update to reduce the Bellman residual? If the answer is yes, then I am not sure that I understood the message out of derivation in pages 3 and 4.

The authors introduce UCB exploration using Q-ensemble and mentioned that they extend the intuition of UCB algorithms in order to propose algorithm 2. But, as the reviewer 2 also touched upon it, I could not find a justification why the variance of k networks, trained using the same procedure but different initialization resembles upper confidence bound. Could you please comment on that?

In light of recent revelations in deep reinforcement learning (i.e. https://arxiv.org/pdf/1709.06560.pdf) and lack of significant improvement of the two proposed methods over DDQN, that would be helpful if the authors could comment about whether they feel their empirical results is an evidence of the significance of their methods.

Notation. In many places in the equations, e.g. first equation in page 2,  the authors used r as a reward, but I guess it should be r(s,a).

Thanks.

---

> ### Author Response · Authors · 2018-01-03
> **reply to clarifications**
>
> We first comment on the significance of the performance improvement. The Double DQN baseline we used is a very fine-tuned baseline, achieving competitive rewards compared with prior published results such as the original Double DQN paper [Van Hasselt, et al. 2016]. In fact, in many games, our results trained with 40-million-frames are already higher than prior results trained with 200-million-frames. For common hyperparameters, our Ensemble Voting and UCB Exploration algorithms used the same as those in our Double DQN implementation.  Both methods achieved significant improved performance over Double DQN. Table 2 in Appendix B shows that Ensemble Voting performs better than Double DQN in 37 out of 49 games evaluated, and UCB Exploration performs better than Double DQN in 38 out of 49 games evaluated. In addition, UCB Exploration performs better than Ensemble Voting in 35 out of 49 games evaluated. We will expand the Results section and include these comparisons.
>
> We do assume that the reward function is deterministic given state and action. We will state out this assumption more clearly in the notations (Section 2.1). We will also replace `r` with `r(s, a)` to make the reward’s dependency on the state and action more clear. We will abbreviate the derivations on pages 3 and 4  and rewrite based on the feedback.
>
> Regarding the upper confidence bound, we use the different network to construct an empirical variance of the estimated Q values for each (s, a) combination. As the Q functions are parametrized by a deep neural network, they do not converge to the same parameters when initialized independently and trained with the same samples due to nonconvexity, thus leading to varied Q value empirical estimates from the Q networks. The uncertainty comes from the variance of the empirical estimates.
>
> We also try constructing the empirical variance from Q networks initialized randomly and trained with samples drawn independently from the replay buffer.  However, this approach does not lead to better performance compared with random initialization only. Thus we conclude the discrepancies in the Q-networks created by independent random initializations contain very useful information for exploration.

---

### Public Comment · (anonymous) · 2017-12-22
**ICLR Reproducibility Challenge Summary**

We attempted to replicate this paper as part of the ICLR Reproducibility Challenge. We built our own implementation of this algorithm on top of OpenAI's existing Double DQN baseline. We attempted to replicate the results on three environments: Space Invaders, Breakout, and UpNDown. In the first two games our results appear to validate the paper's baseline-relative performance, although the specific scores we achieved were quite different. However on the UpNDown environment we were unable to achieve success using their algorithm, doing far worse than the baseline or results in other papers. The cause of our failure to replicate in UpNDown is still unclear. It's plausibly due implementation differences, such as in exactly how Adam was used train the convolutional layers shared by the multiple heads, or whether gradient normalization was in fact used. Based on the score differences between our baselines across all experiments, and since our UCB was implemented on top of our Double DQN baseline, implementation differences between baselines probably also play a role.

The full report is here: https://drive.google.com/file/d/1QVAmKK1ijZkYXeHxdyb6YrP0-K2CXQht/view?usp=sharing
The report appendix includes a link to our codebase.

---

> ### Author Response · Authors · 2018-01-05
> **Thank you for reproducing the results**
>
> We would like to thank you for reproducing and validating the results of our paper. In our implementation, gradients from the multiple heads are first averaged before passing into the gradient update of the convolutional layers.

---

### Public Comment · (anonymous) · 2018-01-02
**ICLR 2018 Reproducibility Challenge**

We reproduce the experiments in the paper "UCB EXPLORATION VIA Q-ENSEMBLES" and verify the main conclusions. Our full report can be found at https://github.com/yifjiang/UCB-review/blob/master/Reproducing%20UCB%20EXPLORATION%20VIA%20Q-ENSEMBLES.pdf. Here is a summary of our work.

The original paper employed the UCB method on bootstrapped DQN and did an experiment on 49 Atari games. We implemented the baseline model, Double DQN, as well as one of the proposed models, UCB Exploration, upon OpenAI baseline models. Due to the constraints on time and computing resources, we attempted to replicate the results on one game (UpNDown). In addition, we also evaluated the models on a simpler environment, CartPole. We got similar results as the original paper on UpNDown. UCB Exploration outperforms Double DQN in this environment. However, the UCB Exploration method does not perform as well as Double DQN in the CartPole environment.

Overall, Our experiments show that the original paper is reproducible. The hyperparameter table provided in the original paper greatly helps the reproduction and improves the soundness of the paper.

---

> ### Author Response · Authors · 2018-01-05
> **Thank you for reproducing the results**
>
> We would like to thank you for reproducing and validating the results of our paper. Regarding the hyperparameter $\lambda$ in UCB exploration, it is set to $\lambda = 0.1$ uniformly for all games evaluated as stated in the middle of Page 5 of the draft. No game-specific fine-tuning was done in the experiments.

---

### Author Response · Authors · 2018-01-05
**Revised manuscript**

Dear reviewers, we have taken your feedback into account and revised the manuscript. A new manuscript has been uploaded.

---

### Decision · Program_Chairs · 2018-01-29
**ICLR 2018 Conference Acceptance Decision**

**Decision:**

Reject

**Comment:**

The idea studied here is interesting, if incremental. The empirical results are not particularly stellar, but it's clear that the authors have done their best to provide reproducible and defensible results. A few sticking points: a) The use of the term 'UCB', as mentioned in an anonymous comment, is somewhat misleading. "Approximate Confidence Interval" might be less controversial; b) there are a number of recent research results on exploration that are worth paying attention to (Plappert et al, O'Donoghue et al.) and worth comparing to, and c) the theoretical results are not always justified or useful (e.g. Equation 9: the bound is trivial, posterior >= 0 or 1).